# Tendencies and attitudes towards dietary supplements use among undergraduate female students in Bangladesh

Ishrat Jahan[1], Abul Bashar Mohammad Neshar Uddin[1], A. S. M. Ali Reza[1,2], Md. Giash Uddin[3], Mohammad Shahadat Hossain[1], Mst. Samima Nasrin[1,2], Talha Bin Emran[4], Md. Atiar Rahman[2]*

1 Department of Pharmacy, International Islamic University Chittagong, Kumira, Chittagong, Bangladesh,
2 Department of Biochemistry and Molecular Biology, University of Chittagong, Chittagong, Bangladesh,
3 Department of Pharmacy, Faculty of Biological Sciences, University of Chittagong, Chittagong, Bangladesh, 4 Department of Pharmacy, BGC Trust University Bangladesh, Chittagong, Bangladesh

* atiar@cu.ac.bd

## Abstract

### Background

Dietary supplements (DS) are products that improve the overall health and well-being of individuals and reduce the risk of disease. Evidence indicates a rising prevalence of the use of these products worldwide especially among the age group 18–23 years.

### Aim

The study investigates the tendencies and attitudes of Bangladeshi undergraduate female students towards dietary supplements (DS).

### Methods

A three-month (March 2018-May 2018) cross-sectional face-to-face survey was conducted in undergraduate female students in Chittagong, Bangladesh using a pre-validated dietary supplement questionnaire. The study was carried among the four private and three public university students of different disciplines in Chittagong to record their prevalent opinions and attitudes toward using DS. The results were documented and analyzed by SPSS version 22.0.

### Results

Ninety two percent (N = 925, 92.0%) of the respondents answered the survey questions. The prevalence of DS use was high in undergraduate female students. The respondents cited general health and well-being (n = 102, 11.0%) and physician recommendation (n = 101, 10.9%) as a reason for DS use. Majority of the students (n = 817, 88.3%) used DS cost monthly between USD 0.12 and USD 5.90. Most of the students (n = 749, 81.0%) agreed on the beneficial effect of DS and a significant portion (n = 493, 53.3%) recommended for a regular use of DS. Highly prevalent use of dietary supplements appeared in Chittagonian

**Data Availability Statement:** All relevant data are within the paper and its Supporting information files.

**Funding:** The author(s) received no specific funding for this work.

**Competing interests:** The authors have declared that no competing interests exist.

undergraduate female students. They were tremendously positive in using DS. The results demonstrate an increasing trend of using DS by the undergraduate females for both nutritional improvement and amelioration from diseases.

## Conclusion

Dietary supplements prevalence was so much higher in students of private universities as compared to students of public universities. Likewise, maximal prevalence is indicated in pharmacy department compared to other departments. Students preferred brand products, had positive opinions and attitudes towards dietary supplements.

## Introduction

Dietary supplement (DS) is a manufactured product containing one or more than one dietary elements such as vitamins, minerals, herbs or botanicals products, fatty acids, proteins purposive to supplement the diet [1]. DS provides nutrients either extracting from food or synthetic sources administered through the mouth as a tablet, capsule, or pill. It also contains other constituents that may not be therapeutically effective or essential for life, such as polyphenols or plant pigments. In many countries, DS's are deliberated as a subset of foods and controlled consequently [2]. DS's are commonly used by adults to progress or maintain health as well as disease recovery [3, 4]. The use of DS is gradually increasing over time over the world [4]. Around 68% of US adults are reported to use DS in different forms. During 2007–2010 in the United States (US), it was assessed that 23% of the supplements was suggested by the health care professional [5]. Another US investigation through NHANES (1999–2012) reported that nearly 52% of responders from the US had an uninterrupted DS consumption which is found as diverse testaments in US and Europe for extensive consumption [6]. Bangladesh is located in South Asia, where the use of DS among individuals is very common. During 2018, an assessment showed that approximately 41.3% of participants used the DS at least once in the previous year. The number of female users was 45%, which was higher than that of male users. Additionally, 33% of the users started taking supplements without any prescription or consultation healthcare professionals such as pharmacists consisting vast knowledge on the safer and effective uses of DS. DS's using rate is less in good health people than in poor health people. The aims, perception and opinions of using DS varies in different age and professional groups while kids are given DS's with the suspicion of nutritional insufficiency [6].

The college and undergraduate students feel they need extra nutrients from DS to compensate their energy deficit for extracurricular activities, study, or part-time jobs. Researchers reported 66% of the five US university students consume DS [7]. 63% of female participants from 16 US medical colleges use DS while male DS consumers were lesser than female [8]. Another study showed that 76.6% of Saudi female students love to use DS [9]. An insignificant difference between male and female DS uses in Japan and 56% of Australian DS using students in 2015 are reported in literature [10]. The micronutrient depletion situation in Bangladesh is growing day by day. In Bangladesh, one-fourth of the population aged 15 to <49 years had chronic energy deficiency or thinness (BMI<18.5) and in Bangladesh, about 25% of the adult population was undernourished or lean (BMI<18.5) aged 46-<60 years [11]. Moreover, females are more prone to suffer from nutritional deficiencies than men due to various factors such as female reproductive biology, low social status, poverty and lack of education [11].

Socio-cultural practices and inequalities in domestic job habits may also increase the possibility of malnutrition for females [12].

In Bangladesh, a group of researches revealed the dietary supplement especially calcium, zinc and iron supplement consumption pattern on pregnant and non-pregnant women [13]. Another research from southern parts of Bangladesh demonstrated that, male and females participants aged between 15–65 years consumed DS without prescriptions [14]. Unfortunately, no such study has been conducted for Bangladeshi female undergraduate students. This research was conducted to determine the prevalence, user opinions and attitudes towards DS among undergraduate female students in Chittagong city of Bangladesh.

## Materials and methods

### Participant consent and ethical consideration

Protocol used in this study for the research work entitled on (Dietary supplement use among undergraduate female students in public and private universities located in Chittagong, Bangladesh: prevalence, opinions, and attitudes) was approved by the institutional ethical review committee (P&D Committee) of International Islamic University Chittagong, Chittagong, Bangladesh-4318 (ref: P&D-147/13-19). Participants were known to purposes, protocol, and importance of the survey. The participation was deliberate except any influence. Before supplying the questionnaire, an informed written consent was searched from students (S1 File).

### Study design and duration

A cross-sectional study was planned and conducted in undergraduate female students for an epoch of three months. All the participants of this study were undergraduate females who had been selected between March 2018—May 2018. The inclusion criteria were: willingness to participate in the study, age >18 years, ability to understand English or Bengali, and participants with previous experience using dietary supplements. Participants who did not match the above criteria were excluded from the study. Finally, incompletely or incorrectly filled questionnaires were also not included.

### Study site

The study was carried out in public and private universities located in Chittagong, Bangladesh. Chittagong division is geographically the largest of the eight administrative divisions of Bangladesh including seven public and six private universities.

### Sample size calculation

Sample size was calculated by using Raosoft, Inc. (USA) [15]. Present survey involved 925 respondents who were undergraduate female students from listed four private and three public universities.

### Study population

This cross-sectional study participants chosen from the Departments of Pharmacy, Bachelor of Business Administration, Economics and Banking, Law, Computer Science and Engineering, Electronics and Electrical Engineering, Quranic Science, Dawa and Islamic Studies, Hadith, English Language and Literature from different universities.

## Purposes of the study

The primary purpose of our survey was to explore the dynamics of using dietary supplements by the undergraduate females, involving the respondent's choice, age, discipline, nature of the institution. The survey was directed among several age ranges of 18–23, 24–26, and 27–30 years. The cause of the ordination is the education period. The inclusion of undergraduate students usually occurs in 18 years, and the duration varies from 4–5 years in different disciplines. The second principal objective was to inquire about DS's differences with the variation of demography, particular opinion, attitudes, disciplines, and ages. Expense regarding dietary supplement usage was recorded and indicated by the BDT (Bangladeshi Taka).

## Research instrument development

The authors prepared a questionnaire with the title of "the dietary supplement questionnaire (DSQ)". It consists of several multiple-choice questions. The DSQ was further polished by a group of specialists led by two assistant professors. A pilot study (100 students) was performed to confirm the questionnaire's reliability and validity using different approaches. We distributed the questionnaire amongst students to get their feedback regarding the understanding and clarity of all questions. The questionnaire was then reviewed by experts in related fields and other expert colleagues within the university. We also asked external reviewers to provide their feedback and opinion in developing/improving the questionnaire to ensure the reliability of the test and compared the results of our pilot study with the results of similar work done previously. We introduced all necessary expert feedback and suggestions accordingly until we had a final questionnaire (S2 File) used in the present study.

The DSQ was subdivided into two parts: Part A contains questions in respect of demographic information like age, study year, ethnic region, current marital status, residence, siblings, use of dietary supplement in the last month, name of the departments, and university type. Part B included the questions where respondents were asked on their major illness, reasons for using DS, types of DS they used, their affordability (cost/month in BDT), adverse reactions experienced from DS, opinions, and attitudes. Furthermore, they were asked about their compliance regarding DS's role in health and if they would personally recommend using DS to others. They were also inquired whether they know about DS they had used or currently using.

## Pilot study and accessibility of the DSQ

A pilot study of the DSQ was performed by distributing the questionnaire among 100 undergraduate female students belonging to different universities' disciplines. A 93% of response rate proved that there were no troubles in understanding the questionnaire. At this point, the DSQ was considered suitable to use.

## Data analysis

The data were analyzed by statistical software Statistical Package for Social Science (SPSS, version 22.0, IBM Corporation, NY). Frequency counts (N) and percentages (%) were introduced to represent the demographic information, opinions, and attitudes. The chi-square ($\chi2$) test, cross tabulation, and p values were used to measure associations between demographic variables and opinions/attitudes. Multinomial logistic regression was conducted to determine different relationships between DS use and demographic characteristics of students.

## Results

The reflection of the respondents on the use of DS has been summarized as follows:

### Demographic information of the respondents

The demographic information of the 925 participants is shown in Table 1. The majority (n = 823, 89.0%), out of 925 respondents, in our study participated from private universities. More than three and a half quarter of the students belonged to the age group 18–23 years (n = 859, 92.9%). The majority of students were in 4th year (n = 343, 37.1%), followed by 3rd year (n = 215, 23.2%), during the study years. Almost all of the respondents of the study were found to be unmarried (n = 885, 95.7%). Majority of the students were living with family (n = 757, 81.8%) and were between 1 and 2 siblings (n = 422, 45.6%). These data were contributed by the department of Pharmacy (25.3%), Bachelor of Business Administration (11%), Economics and Banking (14.4%), Law (14.7%), Computer Science and Engineering (12.8%), Electrical and Electronics Engineering (7.5%), Quranic Science (3.4%), Dawa (1.7%), Hadith (1.3%), English Language and Literature (8%) of different universities shown in Fig 1.

### Suffering of the respondents from major illness

The detailed information for suffering of the respondents from major illness (if there is) is summarized in Table 2. Collectively, most of the students suffer from depression (n = 134, 14.5%) leading to hypertension (n = 96, 10.4%). Besides, less than two-quarters of the students did not have any major illness (n = 399, 43.1%).

**Table 1. Demographic information of the respondents.**

| Items | Subgroups | Sample (N) | Percentage (%) |
|---|---|---|---|
| Age | 18–23 years | 859 | 92.9 |
| | 24–26 years | 66 | 7.1 |
| | 27-30years | - | - |
| Study year | 1st year | 173 | 18.7 |
| | 2nd year | 194 | 21.0 |
| | 3rd year | 215 | 23.2 |
| | 4th year | 343 | 37.1 |
| Ethnic origin | Bangladeshi | 925 | 100.0 |
| | Tribal | - | - |
| | Non-Tribal Bengali | - | - |
| Current marital status | Married | 40 | 4.3 |
| | Unmarried | 885 | 95.7 |
| | Divorced | - | - |
| Residence | Living with family | 757 | 81.8 |
| | Living alone (University accommodation) | 148 | 16.0 |
| | Living alone (Self accommodation) | 20 | 2.2 |
| Siblings | Between 1&2 siblings | 422 | 45.6 |
| | Between 3&5 siblings | 355 | 38.4 |
| | Between 6&8 siblings | 74 | 8.0 |
| | More than 8 siblings | 37 | 4.0 |
| | No siblings | 37 | 4.0 |
| University type | Private | 823 | 89.0 |
| | Public | 102 | 11.0 |

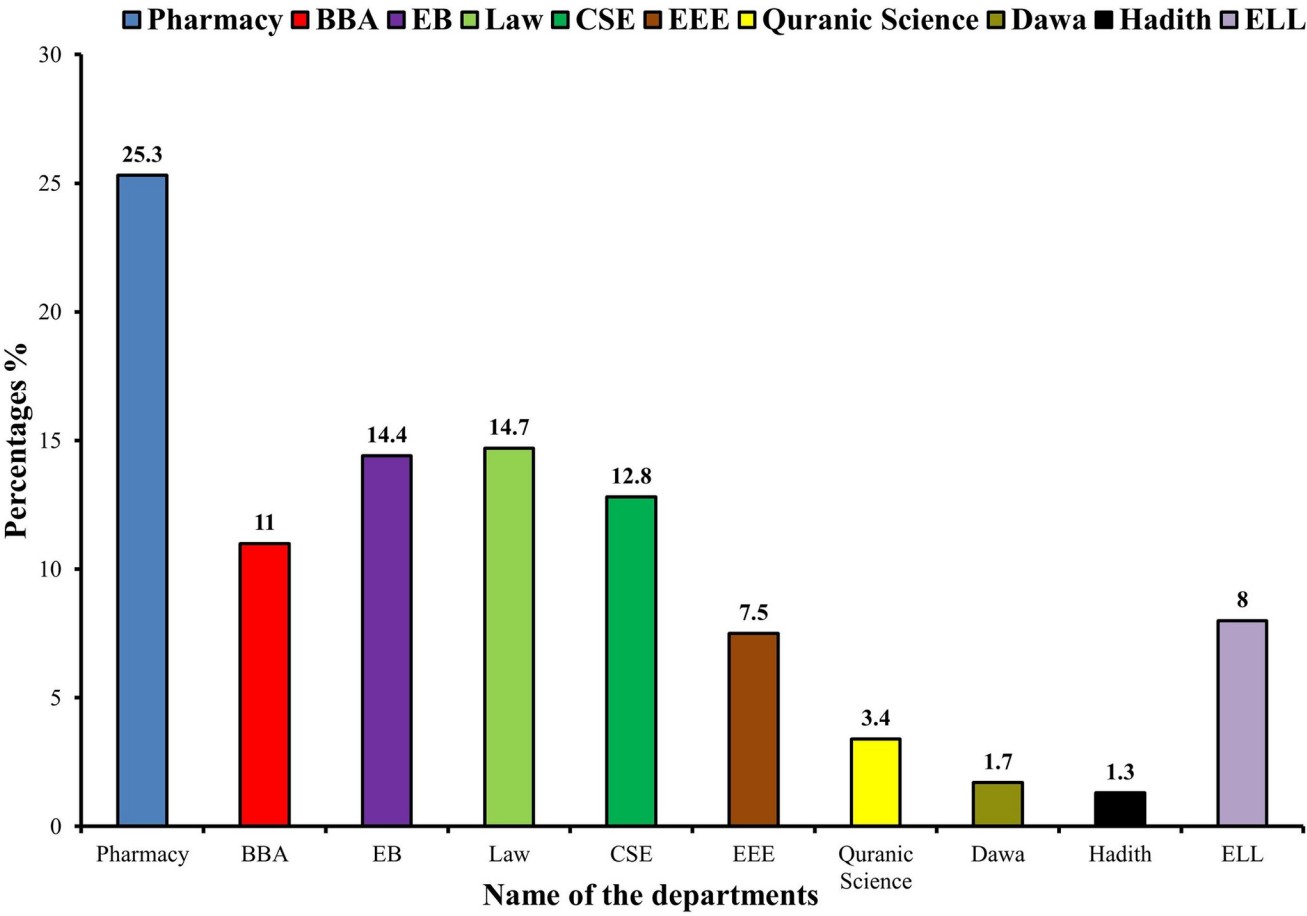

**Fig 1. Percentage of participations from various departmental students.** Here, BBA: Bachelor of Business Administration, EB: Economics and Banking, CSE: Computer Science and Engineering, EEE: Electrical and Electronics Engineering, ELL: English Language and Literature.

**Table 2. Suffering of the respondents from illness (if any).**

| Items | Subgroups | Sample (N) | Percentage (%) |
|---|---|---|---|
| Any major illness | Do not suffer from any illness | 399 | 43.1 |
| Suffer from a major illness | Depression | 134 | 14.5 |
| | Hypertension | 96 | 10.4 |
| | Diabetes mellitus | 38 | 4.1 |
| | Sickle cell anaemia | 54 | 5.8 |
| | Thalassemia | 54 | 5.8 |
| | Asthma | 72 | 7.8 |
| | Rheumatoid arthritis | 36 | 3.9 |
| | Psoriasis | - | - |
| | Ulcer | - | - |
| | Irregular ministration | - | - |
| | Migraine | 21 | 2.3 |
| | Obesity | - | - |
| | Epilepsy | - | - |
| | Others | 21 | 2.3 |

**Table 3. Reasons for use of dietary supplements.**

| Statements | SD, N % | MD, N % | SLD, N % | NA/NOD, N % | SLA, N % | MA, N % | SA, N % |
|---|---|---|---|---|---|---|---|
| Physician recommendations | 123 | 160 | 160 | 180 | 116 | 85 | 101 |
| | 13.3 | 17.3 | 17.3 | 19.5 | 12.5 | 9.2 | 10.9 |
| General health and well being | 119 | 163 | 162 | 180 | 112 | 87 | 102 |
| | 12.9 | 17.6 | 17.5 | 19.5 | 12.1 | 9.4 | 11.0 |
| For weight gaining | 114 | 166 | 166 | 184 | 109 | 88 | 98 |
| | 12.3 | 17.9 | 17.9 | 19.9 | 11.8 | 9.5 | 10.6 |
| For energy source | 122 | 161 | 163 | 181 | 115 | 89 | 94 |
| | 13.2 | 17.4 | 17.6 | 19.6 | 12.4 | 9.6 | 10.2 |
| Immune booster | 119 | 166 | 159 | 178 | 119 | 87 | 97 |
| | 12.9 | 17.9 | 17.2 | 19.2 | 12.9 | 9.4 | 10.5 |
| Increase performance/sports | 121 | 157 | 162 | 178 | 120 | 88 | 99 |
| | 13.1 | 17.0 | 17.5 | 19.2 | 13.0 | 9.5 | 10.7 |
| To control hair fall and skin care | 123 | 156 | 168 | 175 | 116 | 90 | 97 |
| | 13.3 | 16.9 | 18.2 | 18.9 | 12.5 | 9.7 | 10.5 |
| Increase endurance/body building | 121 | 160 | 159 | 180 | 119 | 87 | 99 |
| | 13.1 | 17.3 | 17.2 | 19.5 | 12.9 | 9.3 | 10.7 |
| Memory enhancer | 118 | 162 | 163 | 183 | 113 | 90 | 96 |
| | 12.8 | 17.5 | 17.6 | 19.8 | 12.2 | 9.7 | 10.4 |
| Other reasons(pregnancy-induced anaemia & fatigue) | 120 | 159 | 169 | 180 | 113 | 88 | 96 |
| | 13.0 | 17.2 | 18.3 | 19.5 | 12.2 | 9.5 | 10.4 |
| No reason mentioned | 124 | 158 | 163 | 182 | 112 | 87 | 99 |
| | 13.4 | 17.1 | 17.6 | 19.7 | 12.1 | 9.4 | 10.7 |

(SD = strongly Disagree, MD = Moderately disagree, SLD = Slightly disagree, NA/NOD = Neither agree Nor disagree, SLA = Slightly agree, MA = Moderately Agree, SA = strongly agree).

## Etiologies of using dietary supplements

The detail of the reasons for consuming DS's is presented in Table 3. The greater portion of the respondents took DS to improve their general health and well-being (n = 102, 11.0%) while others consumed with physician's recommendation (n = 101, 10.9%). There was no rationale (n = 124, 13.4%) for the use of DS amongst those who disagreed most strongly.

## Types of dietary supplement used

A mixed-response was received on the type of dietary supplement the respondents sought out. Strongly agreed participants (10.4–10.9%) choose DS to be fortified with whey protein (n = 101, 10.9%), followed by omega 3 fatty acid. Strongly disagreed respondents ranging from 13.0–13.7% who sought DS to receive calcium (n = 127, 13.7%). In contrast, less than two-quarters of the participants disagreed to use no supplement. The details of the types of DS's uses are presented in Table 4.

## Expenditure for dietary supplements per month

A brief summary on monthly cost of DS's is presented in Table 5. The monthly cost of DS's significantly differed among the participants. More than three and a half (n = 817, 88.3%), with DS spent monthly between BDT 10 (USD 0.12) and USD 500 (USD 5.90) participants. BDT 5000 (USD 58.99) and BDT 10 (USD 0.12) were the gross and minimum expenses involved in a month.

**Table 4. Types of dietary supplement used.**

| Statements | SD, N % | MD, N % | SLD, N % | NA/NOD, N % | SLA, N % | MA, N % | SA, N % | Total, N % |
|---|---|---|---|---|---|---|---|---|
| Multivitamins alone or in combination with others | 123 | 158 | 164 | 180 | 118 | 86 | 96 | 925 |
| | 13.3 | 17.1 | 17.7 | 19.5 | 12.8 | 9.3 | 10.4 | 100 |
| Ginseng and Gingko biloba | 120 | 158 | 168 | 180 | 115 | 86 | 98 | 925 |
| | 13.0 | 17.1 | 18.2 | 19.5 | 12.4 | 9.3 | 10.6 | 100 |
| Omega 3 fatty acid | 124 | 162 | 163 | 180 | 113 | 86 | 97 | 925 |
| | 13.4 | 17.5 | 17.6 | 19.5 | 12.2 | 9.3 | 10.5 | 100 |
| Whey protein | 120 | 158 | 166 | 181 | 114 | 85 | 101 | 925 |
| | 13.0 | 17.1 | 17.9 | 19.6 | 12.3 | 9.2 | 10.9 | 100 |
| Calcium | 127 | 161 | 161 | 177 | 113 | 89 | 97 | 925 |
| | 13.7 | 17.4 | 17.4 | 19.1 | 12.2 | 9.6 | 10.5 | 100 |
| Other supplements (prescription and natural products) | 125 | 159 | 168 | 179 | 110 | 86 | 98 | 925 |
| | 13.5 | 17.2 | 18.2 | 19.4 | 11.9 | 9.3 | 10.6 | 100 |
| No supplements used | 362 | 130 | 121 | 121 | 71 | 55 | 65 | 925 |
| | 39.1 | 14.1 | 13.1 | 13.1 | 7.7 | 5.9 | 7.0 | 100 |

(SD = strongly Disagree, MD = Moderately disagree, SLD = Slightly disagree, NA/NOD = Neither agree Nor disagree, SLA = Slightly agree, MA = Moderately Agree, SA = strongly agree).

**Table 5. Cost of dietary supplements per month in Taka.**

| Cost/Month | Sample (N) | Percentage (%) |
|---|---|---|
| BDT 10–500 | 817 | 88.3 |
| BDT 501–1000 | 74 | 8.0 |
| BDT 1001–5000 | 34 | 3.7 |
| BDT 5001–10000 | - | - |

## Adverse reactions experienced from dietary supplements

The details of an adverse reaction are presented in Table 6. The participants were asked if any adverse reactions to the use of DS had occurred. More than 59.0% of the respondents (n = 546) experienced no adverse reactions over DS sue. Nausea, vomiting and diarrhea (n = 75, 8.1%) were the common adverse reactions for those who experienced unwanted effects. Confusion, headaches, and vertigo were encountered by a very insignificant number of respondents (n = 09, 1.0%).

**Table 6. Adverse reactions experienced from dietary supplement.**

| Adverse reactions experienced from dietary supplement | Variables | Sample (N) | Percentage (%) |
|---|---|---|---|
| Did you experience any adverse reactions? | Yes | 295 | 31.9 |
| | No | 546 | 59.0 |
| If yes, what were they? | Nausea, vomiting & diarrhoea | 75 | 8.1 |
| | Confusion, headaches & vertigo | 9 | 1.0 |
| | Hair fall | - | - |
| | Rapid weight gain | - | - |
| | Others | - | - |

**Table 7. Opinions and attitudes of pharmacy students regarding dietary supplement use.**

| Statements | SD, N % | MD, N % | SLD, N % | NA/NOD, N % | SLA, N % | MA, N % | SA, N % | TotalN % |
|---|---|---|---|---|---|---|---|---|
| It prevent chronic illness if used regularly | 105 | 174 | 167 | 186 | 101 | 89 | 103 | 925 |
| | 11.4 | 18.8 | 18.1 | 20.1 | 10.9 | 9.6 | 11.1 | 100 |
| Safe with minimal risk of adverse effects | 109 | 166 | 160 | 184 | 111 | 90 | 105 | 925 |
| | 11.8 | 17.9 | 17.3 | 19.9 | 12.0 | 9.7 | 11.4 | 100 |
| Essential for everyone regardless of age | 102 | 173 | 166 | 185 | 101 | 90 | 108 | 925 |
| | 11.0 | 18.7 | 17.9 | 20 | 10.9 | 9.7 | 11.7 | 100 |
| Prevents cancer | 109 | 166 | 160 | 183 | 109 | 92 | 106 | 925 |
| | 11.8 | 17.9 | 17.3 | 19.8 | 11.8 | 9.9 | 11.5 | 100 |
| Important for health and general well-being | 103 | 168 | 177 | 187 | 97 | 89 | 104 | 925 |
| | 11.1 | 18.2 | 19.1 | 20.2 | 10.5 | 9.6 | 11.2 | 100 |
| Use only as per physician recommendation/ harmful if not used properly | 111 | 166 | 160 | 183 | 113 | 90 | 102 | 925 |
| | 12.0 | 17.9 | 17.3 | 19.8 | 12.2 | 9.7 | 11.0 | 100 |
| Necessary for all ages | 130 | 204 | 153 | 179 | 92 | 82 | 85 | 925 |
| | 14.1 | 22.1 | 16.5 | 19.4 | 9.9 | 8.9 | 9.2 | 100 |

(SD = strongly Disagree, MD = Moderately disagree, SLD = Slightly disagree, NA/NOD = Neither agree Nor disagree, SLA = Slightly agree, MA = Moderately Agree, SA = strongly agree).

## Pharmacy student's opinions and attitudes regarding dietary supplement use

Detailed information on the views and attitudes of pharmacy students about DS usage is provided in Table 7. The data on the responses of pharmacy students using DS were close to each other. More than 14.1% (n = 130) of the respondents strongly disagreed to use DS in all ages while 11.5% (n = 108) strongly agreed on the same regardless of ages Mentionable number of participants agreed to use DS for cancer prevention (n = 106, 11.5%).

## Dietary supplements are good for health

The health specifics of DS use are shown in Table 8. More than three quarters and a half of students accepted that the use of DS is good for health (n = 749, 81.0%). Additionally, less than a quarter of students did not know whether its use is good for health or not (n = 139, 15.0%).

## Personal advice to others when using dietary supplements

Details of personal recommendations relating to the use of DSs by others are given in Table 9. The students were asked about their thoughts concerning DS's use. Almost half of the students (n = 493, 53.3%) recommended the daily use of DS while 37% of doctors did the same.

**Table 8. Dietary supplements are good for health.**

| Items | Sample(N) | Percentage (%) |
|---|---|---|
| Agree | 749 | 81.0 |
| Disagree | 37 | 4.0 |
| Do not know | 139 | 15.0 |

**Table 9. Do you personally recommend use of dietary supplements to others.**

| Items | Sample (N) | Percentage (%) |
|---|---|---|
| Yes, I always recommend | 493 | 53.3 |
| Yes, only when doctors recommend | 342 | 37.0 |
| Not at all | 90 | 9.7 |

**Table 10. Dietary supplement product information.**

| Items | Sample (N) | Percentage (%) |
|---|---|---|
| Brand name | 312 | 33.7 |
| Generic name | 243 | 26.3 |
| Both | 243 | 26.3 |
| Do not know | 127 | 13.7 |

## Dietary supplement product information

The respondents were asked whether they were aware of DS product information. Very significant part of the students (n = 312, 33.7%) recognized the brand name of DS while a smaller fraction sensed either brand or generic name of the product (n = 127, 13.7%). The DS product attributes are shown in Table 10.

## Association of demographic variables with opinions and attitudes

The association of demographic variables with opinions and attitudes are attributed in Table 11. The dependable variable (DV) of student opinion towards DS's use good for health was associated with: age (p-value = 0.927), study year (p-value = 0.677), current marital status (p-value = 0.064), residence (p-value = 0.382), siblings (p-value = 0.832), university type (p-value = 0.735). Moreover, the DV of student opinion towards encouraging the use of DS was associated with: age (p-value = 0.507), study year (p-value = 0.510), current marital status (p-value = 0.490), residence (p-value = 0.097), siblings (p-value = 0.094), university type (p-value = 0.928).

## Multinomial regression analysis

Multinomial regression analysis on detail results is presented in Table 12. Multinomial logistic regression (MLR) was used to interpret the odds ratio for dependent variable (DV) of, "recommending dietary supplement use". And the "age" was considered as an independent variable (IV) where "age variation" was fixed as a categorical IV. Student's attitudes towards recommending DS use were taken as DV. The parameter of IV, i.e. "age: 24–26 years", was redundant and therefore set to zero (0). The regression analysis stated that the odds of, "always recommending DS use" increases with every year-wise increase in age (OR' 0.982). And also Students of "Age: 18-23years" were more likely to "recommend DS use" as compared to students of "Age: 24-26years" (OR' 0.982).

## Discussion

The use of dietary supplements is increasingly growing because they are readily available, cost-effective and consumer-compliant. The current survey was conducted to determine the prevalence, opinions and attitudes towards the use of dietary supplements of undergraduate female students at public and private universities in Chittagong, Bangladesh. The study was prevailed

**Table 11. Association of demographic variables with opinions and attitudes.**

| Sample (N) observed (expected) | | | | | |
|---|---|---|---|---|---|
| **Dietary supplements are good for health** | | | | | |
| **Statements** | **Agree** | **Disagree** | **Do not know** | **χ2** | **p vale** |
| 1. Age (years) | | | | 0.151 | 0.927 |
| 18–23 | 695 (695.6) | 34 (34.4) | 130 (129.1) | | |
| 24–26 | 54 (53.4) | 3 (2.6) | 9 (9.9) | | |
| 2. Study year | | | | 4.001 | 0.677 |
| 1$^{st}$ year | 136 (140.1) | 8 (6.9) | 29 (26) | | |
| 2$^{nd}$ year | 157 (157.1) | 8 (7.8) | 29 (29.2) | | |
| 3$^{rd}$ year | 182 (174.1) | 9 (8.6) | 24 (32.3) | | |
| 4$^{th}$ year | 274 (277.7) | 12 (13.7) | 57 (51.5) | | |
| 3. Ethnic origin | 749 (749) | 37 (37) | 139 (139) | - | - |
| 4. Current marital status | | | | 5.485 | 0.064 |
| Married | 38 (32.4) | 0 (1.6) | 2 (6) | | |
| Unmarried | 711 (716.6) | 37 (35.4) | 137 (133) | | |
| 5. Residence | | | | 4.185 | 0.382 |
| Living with family | 617 (613) | 28 (30.3) | 112 (113.8) | | |
| Living alone (University accommodation) | 119 (119.8) | 7 (5.9) | 22 (22.2) | | |
| Living alone (Self accommodation) | 13 (16.2) | 2 (0.8) | 5 (3.0) | | |
| 6. Siblings | | | | 4.273 | 0.832 |
| Between 1&2 siblings | 341 (341.7) | 16 (16.9) | 65 (63.4) | | |
| Between 3&5 siblings | 285 (287.5) | 13 (14.2) | 57 (53.3) | | |
| Between 6&8 siblings | 61 (59.9) | 4 (3.0) | 9 (11.1) | | |
| More than 8 siblings | 31 (30.0) | 1 (1.5) | 5 (5.6) | | |
| No siblings | 31 (30.0) | 3 (1.5) | 3 (5.6) | | |
| 7. University type | | | | 0.617 | 0.735 |
| Private | 669 (666.4) | 33 (32.9) | 121 (123.7) | | |
| Public | 80 (82.6) | 4 (4.1) | 18 (15.3) | | |
| **Do you encourage the use of dietary supplements?** | | | | | |
| | **Yes, always** | **Yes, only when doctor recommend** | **Not at all** | | |
| 1. Age (years) | | | | 1.359 | 0.507 |
| 18–23 | 454 (457.8) | 322 (317.6) | 83 (83.6) | | |
| 24–26 | 39 (35.2) | 20 (24.4) | 7 (6.4) | | |
| 2. Study year | | | | 5.268 | 0.510 |
| 1$^{st}$ year | 89 (92.2) | 69 (20.2) | 15 (16.8) | | |
| 2$^{nd}$ year | 96 (103.4) | 82 (24) | 16 (18.9) | | |
| 3$^{rd}$ year | 117 (114.6) | 76 (79.5) | 22 (20.9) | | |
| 4$^{th}$ year | 191 (182.8) | 115 (126.8) | 37 (33.4) | | |
| 3. Ethnic origin | 493 (493.0) | 342 (342.0) | 90 (90.0) | - | - |
| 4. Current marital status | | | | 1.428 | 0.490 |
| Married | 25 (21.3) | 12 (14.8) | 3 (3.9) | | |
| Unmarried | 468 (471.7) | 330 (327.2) | 87 (86.1) | | |
| 5. Residence | | | | 7.844 | 0.097 |
| Living with family | 411 (403.5) | 267 (279.9) | 79 (73.7) | | |
| Living alone (University accommodation) | 70 (78.9) | 67 (54.7) | 11 (14.4) | | |
| Living alone (Self accommodation) | 12 (10.7) | 8 (7.4) | 0 (1.9) | | |
| 6. Siblings | | | | 13.575 | 0.094 |
| Between 1&2 siblings | 227 (224.9) | 160 (156.0) | 35 (41.1) | | |

*(Continued)*

**Table 11.** (Continued)

| | | | | | |
|---|---|---|---|---|---|
| Between 3&5 siblings | 183 (189.2) | 135 (131.3) | 37 (34.5) | | |
| Between 6&8 siblings | 41 (39.4) | 29 (27.4) | 4 (7.2) | | |
| More than 8 siblings | 21 (19.7) | 10 (13.7) | 6 (3.6) | | |
| No siblings | 21 (19.7) | 8 (13.7) | 8 (3.6) | | |
| 7. University type | | | | 0.149 | 0.928 |
| Private | 439 (438.6) | 305 (304.3) | 79 (80.1) | | |
| Public | 54 (54.4) | 37 (37.7) | 11 (9.9) | | |

with the Bangladeshi unmarried undergraduate female student ages between 18–23 years. The study showed that DS use has increased in recent years. Several previous studies demonstrated the relationship between higher educational level and DS use. A significant straight relation between higher education level and dietary supplement usage among the full-aged population of France was displayed by Pouchieu and colleagues [16]. Likewise, a survey accomplished by Mileva-Peceva *et al.* indicated that highly educated females consume an extremely significant amount of vitamins and /or mineral food supplements [17]. In another study conducted in 16 medical colleges in the United States, female students used DS's more than male students [8]. In Africa, a study carried among medical students of a Nigerian university reported DS usage by 50% of the students [18]. Therefore, our outcomes are consistent with the above findings.

The majority of the female students, living with families, suffer from major illnesses, i.e., depression, hypertension, diabetes mellitus, sickle cell anemia, thalassemia, asthma, rheumatoid arthritis, migraine, and others. Our results displayed that DS usage is higher in females with high education, which is also coherent with former studies [14]. The National Health and Nutrition Examination Survey (NHANES) 2003–2006 displayed, 61% of DS users passed high-school while only 37% of those were not able to do so [19]. NHANES 1999–2000 reported 62% of DS users with more than high-school education, 48% of those with high-school education and just 35% of those with less than high-school education [20]. Our sample's highest prevalence was in the 18–23 age groups and the minimum in the 24–26 age range. More frequent use of DS's has been observed among female students of private universities. Naqvi and colleagues have also been reported similar observations among female pharmacy undergraduate students from private-sector universities [1]. The socioeconomic condition and facilities may be the reasons behind this. These students typically belong to a relatively loftier socioeconomic condition and are aligned with support programs for students, healthcare facilities, and ease of consultation on physical activity, social and individual problems, and perceived dietary supplements. Whereas most of the students enrolled in the public universities belong to low-socio-economic status, and these universities have low tuition fees [21]. Hence, the prevalence of DS use in public universities is lower than in private universities.

**Table 12.** Multinomial logistic regression analysis.

| Do you personally recommend use of dietary supplements to others? | | Coefficient | Odds ratio (OR) | 95% confidence interval | |
|---|---|---|---|---|---|
| | | | | Lower bound | Upper bound |
| Yes, I always recommend | Age | 1.718 | | | |
| | 18–23 years | −0.018 | 0.982 | 0.425 | 2.269 |
| | 24–26 years | 0[b] | - | - | - |
| Yes, only when doctors recommend | Age | 1.050 | | | |
| | 18–23 years | 0.306 | 1.358 | 0.555 | 3.319 |
| | 24–26 years | 0[b] | - | - | - |

The female students were asked about the reasons behind the use of the DS. Students mentioned various reasons, such as recommendations from physicians, general health and well-being, weight gain, energy source, immune booster, performance or sports increase, hair fall and skincare control, endurance or bodybuilding increase, memory enhancer, other reasons involve pregnancy-induced anemia and fatigue while few others mentioned no reason. In our study, all the mentioned reasons for the usage of DS's were so close to each other. Various DS types include multivitamins alone or in conjunction with Ginseng and Gingko Biloba, omega 3 fatty acid, whey protein, calcium, other supplements (prescription and natural products) non-supplements used by the participants. Previous studies showed that protein supplementation is verily common among students as it improves physical performance [22, 23], whereas omega 3 fatty acid may increase sleeping attributes and decrease depression and anxiety [24]. Abbas *et al*. have reported that anxiety and depression due to academic studies may prevail in undergraduate students [25–27]. Our study indicates that a small portion of the students suffered from adverse reactions by using DS. In most cases, they struggled with nausea, vomiting and diarrhea, which are mostly related to the previous studies reported in the literature [1, 28, 29]. The monthly expense of DS's differed notably among the female students. Most of them (88.3%) indicated a monthly expense between Taka 10–500. A previous study also reported that students spent less than 1 to 3 USD per month [1]. The monetary value of dietary supplements in 2014 amounted to the United States dollar (USD) 165.62 billion and was rising with a compound annual growth rate (CAGR) of 7.3%. The projected market worth is expected to touch USD 278.96 billion by 2021. Asia Pacific region has emerged as the second-largest market for dietary supplements after the US and Canada [30]. Saudi Arabia is the biggest market for dietary supplements in the Middle East region as DS accounts for 4% of total pharmaceuticals sold in the country with an estimated worth of USD 2 billion [1, 31]. Female students preferred brand products rather than generics. Personal preferences are probably the reason behind this. Studies displayed that brand products are trusted to possess preferable quality; therefore, brands may be preferred by the individual [22, 32]. Dwyer and colleagues have reported that poor dietary supplements have resulted in adverse drug events (ADEs) and mortality [33]. Moreover, the quality of pharmaceutical packaging may also affect consumers' preferences. Sabah and colleagues mentioned un-attractive packaging as a reason for the low preference of generic products in Pakistan [34]. Studies report that brands are generally expensive than generics [35]. This study observed pharmacy student's opinions and attitudes regarding DS use. They give various statements from which the most considerable portion of students strongly disagreed with the statement that they only used DS as per the physician's recommendation. Moreover, the largest portion of students strongly agreed that they used DS as essential for everyone regardless of age. All the values of given statements for pharmacy students' opinions and attitudes regarding DS use were very close.

In previous study most of the students also agreed to use DS as per the physician's recommendation [1]. But in another study in Lebanon most of the participants believed that they should use DS's recommendation of friends, peers, and relatives [36]. However, there is an increasing evidence that suggests that dietary supplements can be dangerous if not taken in the prescribed dose [37]. In some cases, over usage of multivitamins may exceed the permissible limits of certain vitamins such as cholecalciferol that may result in adverse effects [33, 38, 39].

This study searched the opinions and attitudes of undergraduate female students about DS. The demographic variables of the age, study year, ethnic origin, current marital status, residence, siblings, and university type were associated. Most of the female students trusted that DS's are good for health. An insignificant number of them disagreed with this, and the remaining students opined that they do not know regarding this. Several studies documented DS users having some positive attitude along with their health [40]. They were more likely to be

prosperous, health sensible, less likely to smoke, and probably consume more hygienic food [41]. Various studies about the demographic features of DS's consumers also revealed that they were chiefly high educated females with young age (20–39 years) [40, 42]. These consequences were compatible with our study outcomes.

## Conclusions

To sum up, the survey provides details about the highly prevalent usage of dietary supplements among undergraduate female students in Chittagong, Bangladesh. The most noteworthy finding of the study was that the prevalence was higher in private universities than students of public universities. The study also found variation in prevalence and attitudes concerning dietary supplement use. The maximal prevalence is shown in health-related departments such as pharmacy compared to other non-health-related departments.

The limitation of our study was the non-inclusion of male students. To understand dietary supplement usage and how opinions change in identical populations according to gender would have determined by including male students. Therefore, it is recommended to lead the study in the male population with a larger sample size to view the impact of participants' demographics such as economic status and other relevant factors such as dietary patterns and lifestyle on dietary supplement consumption.

## Supporting information

**S1 File. Written consent of the respondent.**
(DOCX)

**S2 File. Final questionnaire.**
(DOCX)

**S3 File. Final version of the data sheet with all respondents (925).**
(XLS)

## Acknowledgments

The authors are thankful to the Department of Pharmacy, International Islamic University Chittagong, Bangladesh, for research facilities and other logistic supports. The authors are also thankful to Mr. Mohammed Moniruzzaman Bhuiyan, Department of Statistics, University of Chittagong, Chittagong-4331, Bangladesh, for his generous help in the statistical analysis.

## Author Contributions

**Conceptualization:** A. S. M. Ali Reza, Md. Atiar Rahman.

**Data curation:** Ishrat Jahan, Abul Bashar Mohammad Neshar Uddin.

**Formal analysis:** Ishrat Jahan, A. S. M. Ali Reza.

**Funding acquisition:** Md. Atiar Rahman.

**Investigation:** Abul Bashar Mohammad Neshar Uddin, Mst. Samima Nasrin.

**Methodology:** Mohammad Shahadat Hossain, Mst. Samima Nasrin.

**Project administration:** Md. Atiar Rahman.

**Resources:** Md. Giash Uddin.

**Software:** Md. Giash Uddin, Mohammad Shahadat Hossain, Talha Bin Emran.

**Supervision:** Md. Atiar Rahman.

**Visualization:** Md. Atiar Rahman.

**Writing – original draft:** A. S. M. Ali Reza, Md. Atiar Rahman.

**Writing – review & editing:** Md. Atiar Rahman.

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
