## [Decision Letter · Decision Letter 0]

4 Jan 2021

PONE-D-20-39326

Tendencies and attitudes of young students towards dietary supplements in Bangladesh: female students are more inclined

PLOS ONE

Dear Dr. Rahman,

Thank you for submitting your manuscript to PLOS ONE. After careful consideration, we feel that it has merit but does not fully meet PLOS ONE’s publication criteria as it currently stands. Therefore, we invite you to submit a revised version of the manuscript that addresses the points raised during the review process.

We look forward to receiving your revised manuscript.

Kind regards,

Walid Kamal Abdelbasset, Ph.D.

Academic Editor

PLOS ONE

Journal Requirements:

3.Thank you for including your ethics statement: 

"The survey was permitted to the institutional review committee of International Islamic University Chittagong (ref: P&D-147/13-19).".   

Please amend your current ethics statement to confirm that your named institutional review board or ethics committee specifically approved this study.

4.We note that you have indicated that data from this study are available upon request. PLOS only allows data to be available upon request if there are legal or ethical restrictions on sharing data publicly. For more information on unacceptable data access restrictions, please see http://journals.plos.org/plosone/s/data-availability#loc-unacceptable-data-access-restrictions.

5. Please upload a copy of Figure 1, to which you refer in your text on page 8. If the figure is no longer to be included as part of the submission please remove all reference to it within the text.

Reviewers' comments:

Reviewer's Responses to Questions

**Comments to the Author**

1. Is the manuscript technically sound, and do the data support the conclusions?

Reviewer #1: Yes

Reviewer #2: No

Reviewer #3: Yes

2. Has the statistical analysis been performed appropriately and rigorously? 

Reviewer #1: Yes

Reviewer #2: Yes

Reviewer #3: Yes

3. Have the authors made all data underlying the findings in their manuscript fully available?

Reviewer #1: Yes

Reviewer #2: No

Reviewer #3: No

4. Is the manuscript presented in an intelligible fashion and written in standard English?

Reviewer #1: Yes

Reviewer #2: No

Reviewer #3: Yes

5. Review Comments to the Author

Reviewer #1: Reviewer comments:

Thank you for giving the opportunity to review this article.

The title of the article should be more understandable and self-explanatory.

Please edit the entire manuscript for English grammar and syntax for readability.

Abstract:

1. Background is not clear.

2. Methods: Include the study duration.

3. Avoid abbreviations in the conclusion.

Introduction

1. The introduction part is too short and didn’t mention about important key points.

2. The research question is not discussed and formulated with suitable references.

3. How come this study is differed from reference 7 -9?

4. Mention the aim or objective of the study.

5. Define the clinical significance of this study in related to researchers, clinicians and patients.

Methods

6. Why the search is limited with female students?

7. The duration of search?

8. The selection criteria should be more specific (inclusion and exclusion).

9. How the sample size was calculated - explain clearly?

10. Mention the procedure, reliability and validity of DSQ.

11. Software used for statistical analysis.

Discussion

12. Refine the conclusion according to the objective of your study.

Reviewer #2: Comments are found in the attached file, using side comments/track changes

Major comments:

The title is not representing the content. While all the recruited participants were females, the authors titled the study with "female students are more inclined", more than whom? The authors concluded that health-related departments were more prevalent with DS use, while nothing mentioned in the 12 tables about the academic departments. The study objectives and specific aims are not clearly mentioned. The 12 tables are dissecting different aspects from the mentioned aim of the study. The conclusions are not supported by the presented data. The authors dismissed any article related to DS use in Bangladesh in their discussion.

Reviewer #3: Paper titled (Tendencies and attitudes of young students towards dietary supplements in Bangladesh: female students are more inclined) by Rahman et al., studied the gender difference towards the dietary supplements in a national area in Bangaldish.

I appreciate such simple and hubmble trial, the study is interestingly written with correct statistical analysis. I recommend publication after a minor revision:

If the authors can discuss and metnion the specific type of supplements that are commonly used in this study in reation to the age ranges.

6. PLOS authors have the option to publish the peer review history of their article (what does this mean?). If published, this will include your full peer review and any attached files.

Reviewer #1: No

Reviewer #2: No

Reviewer #3: **Yes: **Sawsan A Zaitone

---

## [Author Response · Author response to Decision Letter 0]

28 Feb 2021

Journal Requirements:

Response: The manuscript has been formatted according to Plos One style.

 Response: The questionnaire has been incorporated as supporting file.

3. Thank you for including your ethics statement: 

"The survey was permitted to the institutional review committee of International Islamic University Chittagong (ref: P&D-147/13-19).". 

Please amend your current ethics statement to confirm that your named institutional review board or ethics committee specifically approved this study.

For additional information about PLOS ONE ethical requirements for human subject’s research, please refer to http://journals.plos.org/plosone/s/submission-guidelines#loc-human-subjects-research.

 Response: The statement has been revised and incorporated in the main text (highlighted in yellow colour).

 Response: All data are available within the revised manuscript. 

 Response: There are no ethical and legal restrictions regarding sharing our data. In our data there is no such information that is sensitive to any particular community. However, no data is left to be shared rather we incorporated all the data produced. The contact information of ethical review committee is Chairman, Ethical Review Committee and Department of Pharmacy, International Islamic University Chittagong, Chittagong, Bangladesh-4318.

 Response: Data are uploaded as supporting file.

5. Please upload a copy of Figure 1, to which you refer in your text on page 8. If the figure is no longer to be included as part of the submission please removes all reference to it within the text.

 Response: The revised figure has been uploaded.

Reviewers' comments:

Reviewer's Responses to Questions

Comments to the Author

1. Is the manuscript technically sounds, and do the data support the conclusions?

Reviewer #1: Yes

Reviewer #2: No

Reviewer #3: Yes

2. Has the statistical analysis been performed appropriately and rigorously?

Reviewer #1: Yes

Reviewer #2: Yes

Reviewer #3: Yes

3. Have the authors made all data underlying the findings in their manuscript fully available?

Reviewer #1: Yes

Reviewer #2: No

Reviewer #3: No

4. Is the manuscript presented in an intelligible fashion and written in Standard English?

Reviewer #1: Yes

Reviewer #2: No

Reviewer #3: Yes

5. Review Comments to the Author

Reviewer #1:

Thank you for giving the opportunity to review this article.

The title of the article should be more understandable and self-explanatory.

Please edit the entire manuscript for English grammar and syntax for readability.

Abstract:

1. Background is not clear.

Response: Authors would like to thank the reviewer for his valuable comment. We have revised the abstract (highlighted in yellow color). 

2. Methods: Include the study duration.

Response: Appreciable query indeed. The duration of the study was three months (March 2018-May 2018), which have been included (highlighted in yellow color).

3. Avoid abbreviations in the conclusion.

Response: Avoided accordingly (highlighted in yellow color).

Introduction

1. The introduction part is too short and didn’t mention about important key points.

Response:We have revised the introduction part. As per reviewer suggestions we also incorporated important key points (highlighted in yellow colour).

2. The research question is not discussed and formulated with suitable references.

Response: The authors specially developed the research question. It consists of several multiple-choice questions. The questionnaire was further polished by a group of specialists led by two assistant professors. We have explained the question formulation method is written in the revised version of the manuscript according to the reviewer's comments (highlighted in yellow colour).

A pilot study (100 students) was performed to confirm the questionnaire's reliability and validity by using different approaches. We distributed the questionnaire amongst students to get their feedback regarding the understanding and clarity of all questions. The questionnaire was then reviewed by experts in related fields and other expert colleagues within the university. We also asked external reviewers to provide their feedback and opinion in developing/improving the questionnaire to ensure the reliability of the test and compared the results of our pilot study with the results of similar work done previously. We introduced all necessary expert feedback and suggestions accordingly until we had a final questionnaire used in the present study.

3. How come this study is differed from reference 7 -9?

Response: Authors are thankful to the reviewer for such a novelty-creating question. By the way, the differences of our research with the cited references are given below:

In reference 7, Lieberman, Harris R., et al. showed dietary supplement use patterns among four United States college students. They considered both male and female respondents. All the respondents received a $10 incentive to complete the survey, and students at Cal State and LSU received a class-based extra credit incentive.

In reference 8 (Spencer, Elsa H et al), the author's main objective was to provide data on medical students' multivitamin and calcium supplement use during medical school. They mostly focused on non-underweight and non-exercising students may be essential targets for messages regarding appropriate and adequate vitamin/mineral use.

In references 9 (Alfawaz et al.), the authors aim to investigate the prevalence of dietary supplement use and its association with sociodemographic/lifestyle characteristics among Saudi female students. Among the participant, 25.5% were married and 74.5% single. Moreover, participants are from undergraduate and graduate level.

In our investigation, the primary focus is to investigate the tendencies and attitudes of Bangladeshi undergraduate female students (age group between 18-23 years) towards dietary supplements.

4. Mention the aim or objective of the study.

Response: The main aim and objective of this study are appended in the abstract section. Also, it has been quoted here.

Dietary supplements are products that improve individuals' overall health and well-being and reduce the risk of disease. Evidence indicates a rising prevalence of these products worldwide, especially among the age group 18-23 years. The study investigates the tendencies and attitudes of Bangladeshi undergraduate female students towards dietary supplements.

5. Define the clinical significance of this study in related to researchers, clinicians and patients.

Response: The micronutrient depletion situation in Bangladesh is growing day by day. In Bangladesh, one-fourth of the population is suffering from chronic energy deficiency. Among them, females are more prone to suffer from nutritional deficiencies than men due to various factors such as female reproductive biology, low social status, poverty, and lack of education. This research will provide comprehensive information related to prevalence, user opinions, and attitudes towards DS among undergraduate female students in Chittagong city of Bangladesh. This will lead the researcher to find out how demographic and lifestyle factors, such as monthly cost, educational status, and lifestyle status, correlate positively with dietary supplements consumption. After all, the clinicians will be able to perceive the research-reflection and they can suggest the said group of females in relevant cases.

Methods

6. Why the search is limited with female students?

Response: Females, particularly in Bangladesh, are more predisposed to suffer from nutritional deficiencies than men due to various factors such as female reproductive biology, low social status, poverty, and lack of education. Socio-cultural practices and inequalities in domestic job habits may also increase the possibility of malnutrition for females. Another reason is also is; no such study has been conducted for Bangladeshi female undergraduate students. Our future goal is to lead similar types of study amidst the male population.

7. The duration of search?

Response: The duration was: March 2018 to May 2018.

8. The selection criteria should be more specific (inclusion and exclusion).

Response: The selection criteria are included in the study design and duration section.

9. How the sample size was calculated - explain clearly?

Response: From seven private and public universities, ten departments were selected by using a lottery. A sample size of 664 was calculated by using Raosoft, Inc. However, by hypothesizing of getting a low number of participants with dietary supplements' previous experience and considering the number of drops out, a total of 1000 undergraduate female students with even ID number was selected randomly from odd and even students ID. Though only 75 female students did not match the study's inclusion criteria, and finally, 925 female students were interviewed and included in the study.

10. Mention the procedure, reliability and validity of DSQ.

Response: We have added the procedure, reliability, and validity of DSQ in the revised manuscript under the Research instrument development section (highlighted in yellow color).

A pilot study (100 students) was performed to confirm the questionnaire's reliability and validity by using different approaches. We distributed the questionnaire amongst students to get their feedback regarding the understanding and clarity of all questions. The questionnaire was then reviewed by experts in related fields and other expert colleagues within the university. We also asked external reviewers to provide their feedback and opinion in developing/improving the questionnaire to ensure the reliability of the test and compared the results of our pilot study with the results of similar work done previously. We introduced all necessary expert feedback and suggestions accordingly until we had a final questionnaire used in the present study.

11. Software used for statistical analysis.

Response: The data were analyzed by statistical software Statistical Package for Social Science (SPSS, version 22.0, IBM Corporation, NY).

Discussion

12. Refine the conclusion according to the objective of your study.

Response: According to reviewer’s remarks, we have revised the entire conclusion section matched with the objective of our study (highlighted in yellow color).

 

Reviewer #2:

Comments are found in the attached file, using side comments/track changes

Response: Comments found in track change are revised as per your suggestions. For further information please check the revised manuscript.

Major comments:

The title is not representing the content. While all the recruited participants were females, the authors titled the study with "female students are more inclined", more than whom? 

Response: Title has been revised.

The authors concluded that health-related departments were more prevalent with DS use, while nothing mentioned in the 12 tables about the academic departments. 

Response: In Table 7, we have summarized the opinions and attitudes of pharmacy students regarding dietary supplement use (highlighted in yellow color).

The study objectives and specific aims are not clearly mentioned. The 12 tables are dissecting different aspects from the mentioned aim of the study. 

Response: According to the reviewers suggestion we have added the objectives and specific aims in the abstract and materials and methods section (highlighted in yellow color).

Aim

The study investigates the tendencies and attitudes of Bangladeshi undergraduate female students towards dietary supplements (DS).

Purposes of the study

The primary purpose of our survey was to explore the dynamics of using dietary supplements by the undergraduate females, involving the respondent's choice, age, discipline, nature of the institution. The survey was directed among several age ranges of 18-23, 24-26, and 27-30 years. The cause of the ordination is the education period. The inclusion of undergraduate students usually occurs in 18 years, and the duration varies from 4-5 years in different disciplines. The second principal objective was to inquire about DS's differences with the variation of demography, particular opinion, attitudes, disciplines, and ages. Expense regarding dietary supplement usage was recorded and indicated by the BDT (Bangladeshi Taka).

The conclusions are not supported by the presented data. The authors dismissed any article related to DS use in Bangladesh in their discussion.

Response: According to reviewers remarked we have revised the entire conclusion section matched with the objective of our study (highlighted in yellow color).

 

Reviewer #3:

Paper titled (Tendencies and attitudes of young students towards dietary supplements in Bangladesh: female students are more inclined) by Rahman et al., studied the gender difference towards the dietary supplements in a national area in Bangladesh.

I appreciate such simple and humble trial, the study is interestingly written with correct statistical analysis. I recommend publication after a minor revision:

If the authors can discuss and mention the specific type of supplements that are commonly used in this study in relation to the age ranges.

Response: Thank you very much for your valuable remarks. The present study investigates the tendencies and attitudes of Bangladeshi undergraduate female students towards dietary supplements. Several classes (Ginseng and Gingko biloba, Omega 3 fatty acid, Whey protein, Calcium, Natural products) supplements have been analysed in this study. The specific type of supplements commonly used in this study has been mentioned in table 4.

 

6. PLOS authors have the option to publish the peer review history of their article (what does this mean?). If published, this will include your full peer review and any attached files.

Do you want your identity to be public for this peer review? For information about this choice, including consent withdrawal, please see our Privacy Policy.

Reviewer #1: No

Reviewer #2: No

Reviewer #3: Yes: Sawsan A Zaitone

---

## [Decision Letter · Decision Letter 1]

8 Mar 2021

PONE-D-20-39326R1

Trends and attitudes towards the use of dietary supplements by female university-students in Bangladesh

PLOS ONE

Dear Dr. Rahman,

Thank you for submitting your manuscript to PLOS ONE. After careful consideration, we feel that it has merit but does not fully meet PLOS ONE’s publication criteria as it currently stands. Therefore, we invite you to submit a revised version of the manuscript that addresses the points raised during the review process.

We look forward to receiving your revised manuscript.

Kind regards,

Walid Kamal Abdelbasset, Ph.D.

Academic Editor

PLOS ONE

Journal Requirements:

Reviewers' comments:

Reviewer's Responses to Questions

**Comments to the Author**

1. If the authors have adequately addressed your comments raised in a previous round of review and you feel that this manuscript is now acceptable for publication, you may indicate that here to bypass the “Comments to the Author” section, enter your conflict of interest statement in the “Confidential to Editor” section, and submit your "Accept" recommendation.

Reviewer #1: All comments have been addressed

Reviewer #3: All comments have been addressed

2. Is the manuscript technically sound, and do the data support the conclusions?

Reviewer #1: Yes

Reviewer #3: Yes

3. Has the statistical analysis been performed appropriately and rigorously? 

Reviewer #1: Yes

Reviewer #3: No

4. Have the authors made all data underlying the findings in their manuscript fully available?

Reviewer #1: Yes

Reviewer #3: No

5. Is the manuscript presented in an intelligible fashion and written in standard English?

Reviewer #1: Yes

Reviewer #3: No

6. Review Comments to the Author

Reviewer #1: Reviewer comments:

Thank you for giving the opportunity to review this article.

Abstract:

1. Mention clearly in the conclusion, what prevalence? DS

Methods

2. Include the selection criteria of participants in detail.

3. Include the reliability and validity of outcome measures with references.

4. Mention who has extracted the data and their experience and qualifications.

Discussion

5. The discussion part should discuss the relation between the DS and its effects with latest references.

6. Add the clinical significance of this article over the participants and researchers.

Reviewer #3: Paper titled (Trends and attitudes towards the use of dietary supplements by female universitystudents in Bangladesh)

Spearman’s rank correlation was employed.Regression analysis was conducted to determine different relationships between DS use and demographic characteristics of students:

This reviewer cannot find the figure in which a regression analysis was done, kindly add to the paper

STat analysis in tables, I find it mandatory to stat analyze the percentages to explore the real differences between groups and conclude better on the results.

7. PLOS authors have the option to publish the peer review history of their article (what does this mean?). If published, this will include your full peer review and any attached files.

Reviewer #1: **Yes: **GOPAL NAMBI

Reviewer #3: No

---

## [Author Response · Author response to Decision Letter 1]

25 Mar 2021

Response to editorial and reviewer’s comments

Dear Editor

Thank you so much for your tremendous effort for having the manuscript reviewed. We have found both the editorial and reviewers comments highly impactful for improving the manuscript. We have addressed all the issues raised by reviewers and editorial office. Please scroll down to see the reviewer’s response below:

Journal Requirements:

Author’s Response: Authors would like to thank the editor for such concern over retracted articles. We, indeed, have not cited any retracted article in our manuscript. Additionally, the manuscript has been formatted according to Plos One style. 

Comments to the Author

1. If the authors have adequately addressed your comments raised in a previous round of review and you feel that this manuscript is now acceptable for publication, you may indicate that here to bypass the “Comments to the Author” section, enter your conflict of interest statement in the “Confidential to Editor” section, and submit your "Accept" recommendation.

Reviewer #1: All comments have been addressed

Reviewer #3: All comments have been addressed

2. Is the manuscript technically sound and do the data support the conclusions?

Reviewer #1: Yes

Reviewer #3: Yes

3. Has the statistical analysis been performed appropriately and rigorously?

Reviewer #1: Yes

Reviewer #3: No

Author’s Response: Authors have rechecked the probable inadequacy of statistical analysis and addressed in a right manner.

4. Have the authors made all data underlying the findings in their manuscript fully available?

Reviewer #1: Yes

Reviewer #3: No

Author’s Response: In response to the reviewer 3, authors would like to show the gratitude to the reviewer for his keen observation. By the way, authors have ensured to make all the data available in the main manuscript and supporting files of the revised manuscript. 

5. Is the manuscript presented in an intelligible fashion and written in standard English?

Reviewer #1: Yes

Reviewer #3: No

Author’s Response: Authors are happy to denounce that the manuscript has been crosschecked by all the authors for any grammatical or linguistic error. However, authors will appreciate very particular suggestion or recommendation for any mistake/change.

6. Review Comments to the Author

Reviewer #1: Reviewer comments:

Thank you for giving the opportunity to review this article.

Abstract:

1. Mention clearly in the conclusion, what prevalence? DS

Author’s Response: The sentence has been rephrased accordingly. 

Methods

2. Include the selection criteria of participants in detail.

Author’s Response: Authors wish to thank the reviewer for insightful suggestion. The inclusion criteria for participants were: 

i. Undergraduate female students; 

ii. Willingness to participate in the study, 

iii. Age >18 years; 

iv. Ability to understand English or Bengali, and 

v. Previous experiences for dietary supplements.

3. Include the reliability and validity of outcome measures with references.

Author’s Response: Very appreciating comments indeed from the reviewer. We have included the reliability and validity of outcome measures in the revised manuscript under the Research instrument development section (highlighted in yellow color). We have supplemented the final questionnaire for your kind consideration. 

A pilot study (100 students) was performed to confirm the questionnaire's reliability and validity using different approaches. We distributed the questionnaire amongst students to get their feedback regarding the understanding and clarity of all questions. The questionnaire was then reviewed by experts in related fields and other expert colleagues within the university. We also asked external reviewers to provide their feedback and opinion in developing/improving the questionnaire to ensure the reliability of the test and compared the results of our pilot study with the results of similar work done previously. We introduced all necessary expert feedback and suggestions accordingly until we had a final questionnaire used in the present study.

4. Mention who has extracted the data and their experience and qualifications.

Author’s Response: Data were extracted by the well-trained final year B. Pharm/M.Pharm research fellows under the direct supervision of a Professor of Biochemistry, working since 16 years (PD from Japan), who has a long expertise on the functional food and alternative medicine filed. Before data collection, they have attended seven days of hands-on training on the dietary supplement questionnaire (DSQ). The whole research work was assisted by two Assistant Professors (having eight to ten years research experience) of the department of pharmacy, International Islamic University Chittagong, Bangladesh.

Discussion

5. The discussion part should discuss the relation between the DS and its effects with latest references. 

Author’s Response: Very thoughtful suggestion, thank you so much dear reviewer. Discussion part has been revised very carefully in accord with the suggestion of the reviewer. The relation between DS and its effects has been well-discussed and changes of the revised manuscript in this regard have been have been highlighted using a yellow text-highlighter (…..).

6. Add the clinical significance of this article over the participants and researchers.

Author’s Response: We appreciate such a reader-demanded suggestion from the reviewer. Indeed, the status of micronutrient depletion in Bangladesh is growing over the days. In Bangladesh, one-fourth of the population has been suffering from chronic energy deficiency. Among them, females are more prone to suffer from nutritional deficiencies than male because of various factors such as lack of awareness among females, their abnormalities in reproductive biology, lack of adequate knowledge, social and economic discrimination in prioritization for females. This research will create an extensive observation and perception on the prevalence of dietary supplements, necessity and trending of supplements by the educated females, their opinions over experiences and overall attitudes towards the use of DS among undergraduate female students in Chittagong, Bangladesh. This will lead the researcher to determine how demographic and lifestyle issues such as monthly cost, educational status and life-status expenditure are correlated with dietary supplement consumption.

Reviewer #3: Paper titled (Trends and attitudes towards the use of dietary supplements by female university students in Bangladesh)

Spearman’s rank correlation was employed. Regression analysis was conducted to determine different relationships between DS use and demographic characteristics of students:

This reviewer cannot find the figure in which a regression analysis was done, kindly add to the paper

Stat analysis in tables, I find it mandatory to stat analyze the percentages to explore the real differences between groups and conclude better on the results.

Spearman’s rank correlation was employed. 

Author’s Response: Thank you for your worthy remarks. We are confirming through several discussions with our senior statistician that chi-square test is the best befitting analytical tool for our study and we accomplished that to interpret the associations between demographic information and opinions/attitudes our study. We are truly sorry and apologizing for misdirecting the analytical approach on stating Spearman's rank correlation which was not included indeed. We hope the reviewers will accept the author’s clarification. We are again thanking the reviewer for keen observation.

Regression analysis was conducted to determine different relationships between DS use and demographic characteristics of students. This reviewer cannot find the figure in which a regression analysis was done, kindly add to the paper.

Author’s Response: Many thanks for this comment to improve the quality of our manuscript. In our study, we actually performed the multinomial logistic regression, which is relevant for this study. The findings of the multinomial logistic regression analysis are presented in Table 12. Addition/changes in the revised manuscript according to the reviewer's comments have been highlighted in the text using a yellow color highlighter (…..).

Stat analysis in tables, I find it mandatory to stat analyze the percentages to explore the real differences between groups and conclude better on the results.

Author’s Response: Thank you for your comments. In our study's statistical analysis, the percentages to explore the real differences between groups were performed and presented in Tables 1 to 10. Addition/changes in the revised manuscript according to the reviewer's comments have been highlighted in the text using a yellow color highlighter (…..).

---

## [Decision Letter · Decision Letter 2]

29 Mar 2021

Trends and attitudes towards the use of dietary supplements by female university-students in Bangladesh

PONE-D-20-39326R2

Dear Dr. Rahman,

We’re pleased to inform you that your manuscript has been judged scientifically suitable for publication and will be formally accepted for publication once it meets all outstanding technical requirements.

Kind regards,

Walid Kamal Abdelbasset, Ph.D.

Academic Editor

PLOS ONE

Additional Editor Comments (optional):

Reviewers' comments:

Reviewer's Responses to Questions

**Comments to the Author**

1. If the authors have adequately addressed your comments raised in a previous round of review and you feel that this manuscript is now acceptable for publication, you may indicate that here to bypass the “Comments to the Author” section, enter your conflict of interest statement in the “Confidential to Editor” section, and submit your "Accept" recommendation.

Reviewer #1: All comments have been addressed

Reviewer #3: All comments have been addressed

2. Is the manuscript technically sound, and do the data support the conclusions?

Reviewer #1: Yes

Reviewer #3: Yes

3. Has the statistical analysis been performed appropriately and rigorously? 

Reviewer #1: Yes

Reviewer #3: Yes

4. Have the authors made all data underlying the findings in their manuscript fully available?

Reviewer #1: Yes

Reviewer #3: No

5. Is the manuscript presented in an intelligible fashion and written in standard English?

Reviewer #1: Yes

Reviewer #3: Yes

6. Review Comments to the Author

Reviewer #1: The authors have satsfactorily justified the comments raised by me and the article can be published in the present format.

The authors have satsfactorily justified the comments raised by me and the article can be published in the present format.

Reviewer #3: The revised form of paper titled (Trends and attitudes towards the use of dietary supplements by female university students in Bangladesh) by Md. Atiar Rahman et al. is improved compared to the original one

Thanks for the authors for addressing the reviewer's comments.

7. PLOS authors have the option to publish the peer review history of their article (what does this mean?). If published, this will include your full peer review and any attached files.

Reviewer #1: **Yes: **Gopal Nambi

Reviewer #3: **Yes: **Sawsan A. Zaitone

---

## [Editor Report · Acceptance letter]

1 Apr 2021

PONE-D-20-39326R2 

Tendencies and attitudes towards dietary supplements use among undergraduate female students in Bangladesh 

Dear Dr. Rahman:

I'm pleased to inform you that your manuscript has been deemed suitable for publication in PLOS ONE. Congratulations! Your manuscript is now with our production department. 

Kind regards, 

on behalf of

Dr. Walid Kamal Abdelbasset 

Academic Editor

PLOS ONE